# Liquid Biopsy in Hepatocellular Carcinoma: Opportunities and Challenges for Immunotherapy

**DOI:** 10.3390/cancers13174334

**Published:** 2021-08-27

**Authors:** Panagiota Maravelia, Daniela Nascimento Silva, Giulia Rovesti, Michael Chrobok, Per Stål, Yong-Chen Lu, Anna Pasetto

**Affiliations:** 1Department of Laboratory Medicine Karolinska Institutet, 14152 Stockholm, Sweden; daniela.silva@ki.se (D.N.S.); giulia.rovesti@ki.se (G.R.); Michael.Chrobok@ki.se (M.C.); 2Division of Oncology, Department of Medical and Surgical Sciences for Children & Adults, University-Hospital of Modena and Reggio Emilia, 41100 Modena, Italy; 3Unit of Gastroenterology and Hepatology, Department of Medicine/Huddinge, Karolinska Institutet, Department of Upper GI Diseases, Karolinska University Hospital, 14186 Stockholm, Sweden; per.stal@ki.se; 4Department of Pathology, Winthrop P. Rockefeller Cancer Institute, University of Arkansas for Medical Sciences, Little Rock, AR 72205, USA; YLu@uams.edu

**Keywords:** hepatocellular carcinoma (HCC), immunotherapies, liquid biopsy, circulating tumor DNA (ctDNA), circulating tumor cells (CTC)

## Abstract

**Simple Summary:**

Hepatocellular carcinoma (HCC) causes many deaths worldwide, and current treatments have limitations. Immunotherapies have shown the most promising clinical outcomes for advanced HCC. However, there are many patients with HCC who still respond poorly to these treatments. Circulating biomarkers that can easily be obtained through blood sampling are promising in predicting treatment responses, since they are minimally invasive and enable us to constantly monitor disease progression. The aim of this review is to discuss the most promising types of blood-based biomarkers for the diagnosis and prognosis of HCC, with the focus on circulating tumor cells and circulating tumor DNA. We also discuss technologies for detecting these biomarkers, as well as their clinical applications for immunotherapies in HCC. We conclude that despite their encouraging results to accurately predict responses to immunotherapies, more and larger clinical studies are still necessary, in order to improve the precision of biomarkers, which are used in the treatment decision for patients with HCC.

**Abstract:**

Hepatocellular carcinoma (HCC) is one of the deadliest cancer types worldwide. HCC is often diagnosed at a late stage when the therapeutic options are very limited. However, even at the earlier stages, the best treatment is liver transplantation, surgical resection or ablation. Surgical resection and ablation may carry a high risk of tumor recurrence. The recent introduction of immunotherapies resulted in clinical responses for a subgroup of patients, but there were still no effective predictive markers for response to immunotherapy or for recurrence after surgical therapy. The identification of biomarkers that could correlate and predict response or recurrence would require close monitoring of the patients throughout and after the completion of treatment. However, this would not be performed efficiently by repeated and invasive tissue biopsies. A better approach would be to use liquid biopsies including circulating tumor DNA (ctDNA), circulating RNA (e.g., microRNAs), circulating tumor cells (CTC) and extracellular vesicles (EVs) (e.g., exosomes) for disease monitoring in a non-invasive manner. In this review, we discuss the currently available technology that can enable the use of liquid biopsy as a diagnostic and prognostic tool. Moreover, we discuss the opportunities and challenges of the clinical application of liquid biopsy for immunotherapy of HCC.

## 1. Introduction

It has been estimated that more than one million deaths will be attributed to hepatocellular carcinoma (HCC) by 2030 [1], making it one of the deadliest cancer types worldwide. Depending on the stage of HCC, the treatment options can vary. When diagnosed at an early stage, the standard of care options include resection, local ablation or liver transplantation, but the risk of tumor recurrence still remains high [2]. When diagnosed at an intermediate stage, treatment options are limited to transarterial chemoembolization, whereas systemic therapies, such as the multi-kinase inhibitors sorafenib or lenvatinib, have, until recently, been the treatment of choice for late-stage tumors [3,4]. For advanced HCC, as for a few other solid cancers, immunotherapy is one of the most promising and novel treatment approaches. A number of ongoing clinical trials have been reported [2,5] in which various immunotherapies, such as immune checkpoint inhibitors (ICIs), are utilized for the treatment of HCC, either alone or in combination with targeted and/or systemic therapies [2,5].

Despite the great clinical benefit that immunotherapies have offered, there are still many patients who do not respond or respond poorly to this type of treatment. In particular, only about 15–20% of advanced HCC patients respond to ICIs [6]. The reasons for the unsatisfactory clinical responses are not clear. One major area of research is indeed focusing on the identification of biomarkers that can better predict tumor responses to the immunotherapy, in order to improve the clinical outcomes and cover a broader number of cancer patients. Various biomarkers have been shown to predict responses to ICIs including tumoral PD-L1 expression [7] and tumor mutational burden (TMB) [8]. A higher TMB, based on genomic profiling of various tumor biopsies, may reflect a higher likelihood for response to ICIs [9], whereas PD-L1 expression can positively correlate with better responses to anti-PD-L1 therapy [10,11]. In addition, gene expression analysis on HCC adjacent tumor tissues has been able to identify signatures correlated with improved survival [12]. However, tissue biopsies require invasive tumor sampling, therefore making it harder to collect multiple samples and comprehensively track tumor genomic changes throughout the treatment [13,14,15]. In particular, HCC is a heterogeneous and molecularly complex cancer type, and conventional tissue biopsies are not able to fully reflect its heterogeneity and thus accurately predict therapy efficacy [16]. In addition, unlike other solid tumors, tissue biopsies for HCC are not frequently available, since diagnosis mainly relies on imaging [17]. Additionally, at a late stage, when the lesions are unresectable, a liver biopsy is usually not recommended for advanced HCC [18], while there is a risk of extrahepatic tumor spread along the needle track [19].

Liquid biopsy, where only a blood sample is taken to analyze circulating tumor cells (CTC) [20] or circulating tumor DNA (ctDNA) [21], can overcome these issues due to its minimally invasive nature. Additionally, it can be used to monitor the disease status systematically [22]. Alpha-fetoprotein (AFP) is one of the first liquid biopsy biomarkers used for the early diagnosis of HCC [23]. However, concerns about its sensitivity and high levels of AFP in non-HCC patients highlight the need to identify more sensitive and reliable biomarkers, which can be used alone or in combination with AFP [16].

In this review, we will describe the currently available technology that can enable the use of liquid biopsy, with the focus on CTCs and ctDNA as diagnostic and prognostic tools. Furthermore, we will discuss the opportunities and challenges of the clinical application of liquid biopsy for immunotherapy of HCC.

## 2. Liquid Biopsy in HCC

Liquid biopsy refers to all the non-solid biologic materials used for the diagnosis and monitoring of HCC and is mainly based on the detection of ctDNA [24,25,26], circulating RNAs (e.g., microRNAs) [27,28,29], CTCs [20,30,31] and extracellular vesicles (EVs) (e.g., exosomes) [32,33,34] (Figure 1). In the following paragraphs, we will focus on two of the most promising liquid biopsy types in HCC, ctDNA and CTCs.

### 2.1. ctDNA in HCC

ctDNA can arise in the bloodstream of cancer patients as a result of tumor cell apoptosis or necrosis [35]. As ctDNA represents the total tumor genome, its role in determining clinical outcomes gains more and more attention, especially in cases of advanced and unresectable HCC in which surgical or other invasive procedures, including tissue biopsy, are not recommended [18]. ctDNA contains cancer-associated molecular characteristics, such as mutational signatures [36], epigenetic changes [37,38] and cancer-derived viral sequences [39], which allow its discrimination from total normal circulating cellular free DNA [40,41,42]. Therefore, it could significantly contribute to the improvement in sensitivity for the current diagnostic tools, such as AFP, whose sensitivity remains at an average of 50% among HCC cases [16]. In a study including 42 patients with unresectable primary liver cancer, ctDNA could correlate more closely with the tumor load and could predict treatment efficacy with higher sensitivity, compared to AFP or imaging [18]. Another pilot study showed that ultra-deep targeted sequencing of cell-free DNA (cfDNA) could confidently detect somatic mutations, which were previously identified in tissue biopsies and were frequently found in HCC patients, highlighting the benefits of cfDNA-derived mutation sequencing [43]. Similarly, in a larger cohort study enrolling 121 advanced HCC patients, mutation profiling of ctDNA revealed mutations in the most frequent HCC-associated driver oncogenes and tumor suppressors, including the TERT promoter, TP53, CTNNB1, PTEN, AXIN1, ARID2, KMT2D and TSC2. This technique was able to reveal predictive mutational signatures associated with responses to systemic therapy with tyrosine kinase inhibitors (TKIs) [36]. Other studies based on detection and mutational analysis of ctDNA also showed promising results [40,44,45,46,47].

Despite the proven valuable role of ctDNA as a tumor biomarker, it still has some limitations, including the low levels of detection in the early stages, which makes it challenging for the early diagnosis of HCC. Another limitation is the lack of standardized procedures to prepare samples and analyze data [16]. Lastly, this approach is limited by the uncertain ability to capture spatial tumor heterogeneity, which can reflect clonal differences within or across cancer metastatic sites [48]. This implies that combinational and/or multi-parametric approaches may be needed, in order to increase the sensitivity and specificity of ctDNA as a biomarker for HCC.

### 2.2. CTCs in HCC

CTCs are also emerging as a promising biomarker for the prediction of HCC treatment efficacies. CTCs arise in the circulation after detachment from primary or metastatic tumor lesions [49]. They differ from other types of cancer biomarkers as they represent viable tumor cells circulating in the patient’s bloodstream. Therefore, CTCs can also provide comprehensive genetic information about tumor heterogeneity and drug sensitivity [20]. CTCs have been approved by the FDA as diagnostic markers for specific epithelial cancers [16]. However, their diagnostic role in HCC still requires further studies. A widely known CTC biomarker is the epithelial cell adhesion molecule (EpCAM) [49], a pan-cancer biomarker which has also been observed in HCC patients [50]. Several studies highlighted the role of EpCAM+ CTCs in predicting HCC recurrence after surgery, as well as their associations with disease progression, vascular invasion and overall survival [51]. Detection of EpCAM-positive CTCs with co-existing T regulatory cells (CD4+/CD25+/Foxp3+) indicated HCC recurrence [52]. In this study, patients with high CTC/Tregs levels had a significantly higher risk of developing postoperative HCC recurrence than those with low CTC and Treg levels. In addition, other subtypes of CTCs were also explored, including the presence of mesenchymal CTCs, which were also associated with a higher risk of tumor recurrence in HCC patients [53,54].

Despite the highly promising role of CTCs as a biomarker for HCC [20], it remains challenging to detect HCC CTCs early and accurately because of the lack of specific markers. Another limitation is that the frequencies of CTCs are usually low in the circulation, especially at the early stages [16,20]. Thus, combinational strategies may be needed, in order to improve the prognostic and diagnostic value of HCC. We will discuss this in detail in Section 4.

## 3. Technology Platforms for Isolation and Detection

### 3.1. The Detection of CTCs


**(i)** 
**Biophysics-based approach**



The first approach utilizes the difference in biophysical properties between CTCs and normal blood cells, such as size and density. For example, the ISET (isolation by size of epithelial tumor cells) technique utilizes a polycarbonate membrane with calibrated, 8 mm-diameter, cylindrical pores to filter out rare CTCs from the blood [55]. After filtering, each membrane is allowed to dry and then stained with H&E or antibodies. In addition, fluorescence in situ hybridization (FISH) and PCR-based genetic analyses can be applied to ISET-isolated cells. The ISET technique has been used to isolate CTCs from HCC patients [55,56].

Another approach is based on the difference in density. Ficoll-Paque was originally developed to isolate peripheral blood mononuclear cells (PBMCs) from other blood components. Ficoll-Paque is placed at the bottom of a conical tube, and then the blood is layered above Ficoll-Paque. After centrifugation, PBMCs are located in a layer between the plasma and Ficoll-Paque. The same approach can be used to enrich CTCs from other blood cells [57]. By following the same process, CTCs are enriched in the PBMC layer. In general, biophysics-based approaches can easily enrich CTCs, but the purity is far less than the antibody-based approach. Therefore, additional steps are required to detect CTCs, such as antibody staining or PCR-based approaches [57,58,59] (Figure 2Ai).


**(ii)** 
**Antibody-based approach**



The most common approach is using an antibody to detect the cell surface marker EpCAM on CTCs. EpCAM is over-expressed on tumor cells, but it can be expressed on some normal epithelial cells [60]. CellSearch is the first FDA-cleared test to provide CTC-related information to clinicians [61,62]. In a CellSearch system, CTCs in a plasma sample are enriched by EpCAM antibody-labeled magnetic beads. These cells are further stained with fluorescence-labeled CD45, cytokeratin-8 (CK-8), CK-18 and CK-19 antibodies. Cells with CD45(−), CK-8(+), CK-18(+) and CK-19(+ or −) are considered as CTCs. CellSearch has been cleared for use as a diagnostic test for patients with metastatic breast, prostate or colorectal cancer, in conjunction with other diagnostic tests. This system has also been used for several research studies in HCC [31,63,64]. In addition to CellSearch, other commonly used techniques, such as MACS (magnetic-activated cell sorting), have also been used to enrich antibody-labeled CTCs [65,66,67] (Figure 2Aii).

### 3.2. The Detection of ctDNA


**(i)** 
**Quantitative PCR-based approach**



The quantitative PCR-based approach utilizes pre-designed PCR primer/probe sets to detect known mutations, usually hotspot mutations, in the plasma DNA. In recent years, scientists have employed droplet digital PCR (ddPCR) techniques to further improve the precision [68]. ddPCR is very similar to quantitative PCR, except that one or a few DNA templates are compartmentalized by small droplets. Therefore, DNA templates can be amplified independently without interference, in order to achieve a better precision. In addition, ddPCR overcomes issues of normalization to housekeeping since it is based on absolute quantification of sample fractionalizing and statistics correction for multiple target molecules identified per droplet [69]. In an HCC study, ddPCR could detect hotspot mutations in plasma samples from 48 HCC patients [70]. In this study, ddPCR assays were performed to target TP53 (c.747G > T), CTNNB1 (c.121A > G, c.133T > C) and TERT (c.1-124C > T) using wild-type and mutant probes. A total of 56.3% of the HCC patients in the study were found to have at least one of these mutations in ctDNA (Figure 2Bi).


**(ii)** 
**NGS-based approach**



The NGS (next-generation sequencing) technique can be utilized to detect tumor DNA from patients’ plasma, by either detecting tumor mutations or chromosome abnormalities. The advantage of this approach is that it can detect tumor DNA in plasma without prior knowledge or pre-defined hotspot sites. However, because the vast majority of plasma DNA comes from normal cells, it can be technically challenging to conduct this.

Tumor cells usually have some chromosomal alterations, including copy number changes, amplifications, deletions and translocations, which can be detected by low-coverage whole-genome sequencing, followed by bioinformatics analysis [71]. The sensitivity and specificity of this approach largely rely on the depths of sequencing and also the design of bioinformatic pipelines. Additionally, the nature of tumor cells, such as the magnitude of chromosomal alterations, can also directly affect the sensitivity of this approach. This approach has been used in many cancer types, including HCC. For example, shortened DNA associated with copy number aberrations was detected in 84% of HCC patients in one report [72].

A more sensitive approach is to detect tumor mutations directly by NGS. For instance, an approach to detect tumor mutations from plasma, called CancerSEEK, has been developed [21]. Plasma samples were collected from 1005 patients with nonmetastatic, clinically detected common cancers, including HCC. DNA materials were purified from plasma and then amplified by 61 PCR primer pairs targeting regions of interest from 16 genes, including TP53, KRAS, CTNNB1, PIK3CA, APC, EGFR, PTEN, FGFR2, CDKN2A, GNAS, PPP2R1A, AKT1, HRAS, BRAF, FBXW7 and NRAS. PCR products were deep sequenced, and hotspot mutations were identified by bioinformatic analysis. The median sensitivity of CancerSEEK was 70% among eight cancer types, and 97% for HCC. The specificity for CancerSEEK was greater than 99%.

Although NGS approaches could detect ctDNA in patients with some tumor burdens, the sensitivity was not sufficient to detect residual diseases for patients with nonmetastatic cancer. A targeted digital sequencing (TARDIS) approach was developed to improve the sensitivity [73]. Tumor mutations from each individual breast cancer patient were identified by whole-exome sequencing using their tumor biopsy specimens. Approximately 18 patient-specific mutations were selected for PCR amplification for each patient. Importantly, a random nucleotide sequence, also known as a unique molecular identifier (UMI), was added to each single-stranded DNA template prior to the PCR amplification. The PCR products were deep sequenced, and the UMI and fragment size were used to remove potential errors introduced during the PCR amplification. This approach could improve the sensitivity up to 100-fold, compared to other approaches. As a result, ctDNA was detected in 100% of patients with early and locally advanced breast cancer prior to treatments, and in 12 out of 13 patients with invasive or in situ residual diseases after treatments (Figure 2Bii).

## 4. Liquid Biopsy as a Diagnostic and Prognostic Tool

Historically, serum AFP and diagnostic imaging have been the primary diagnostic modalities used for HCC [74]. Elevated levels of AFP have been associated with increased tumor size and portal vein thrombosis, as well as an increased risk of recurrence after liver transplantation [51,75]. However, the role of AFP as a biomarker has a limited diagnostic value because of the low sensitivity in HCC, at 50% [51]. Alternatively, other markers such as the AFP lectin fraction (AFP-L3) and des-y-carboxy prothrombin (DCP) have been shown to improve the diagnostic performances when used in combination with AFP [76]. Besides these, Glypican 3 (GPC3) [77], cytokeratin 19 (CK19) [78], Golgi protein 73 (GP73) [79], osteopontin [80], squamous cell carcinoma antigen (SCCA) [81] and annexin A2 [82] have all been shown to have diagnostic and prognostic roles in HCC as well, but they have not been validated sufficiently for routine clinical use [20].

Therefore, research on biomarker combinations has been performed in order to provide more accurate and valuable information for a future individualized HCC diagnosis and/or prognosis assessment [83]. In this context, liquid biopsy has been explored as a way to monitor cancer prognosis and diagnosis in a non-invasive manner. This technology has shown promising results in early diagnosis [84], detection of minimal residual disease [85] and decision making for systemic therapies of different types of cancers, including HCC [8,43,86].

Among all liquid biopsy analytes, ctDNA plays an important role in HCC prognosis [17]. ctDNA maintains the same genomic signatures that are present in the matching tumor tissue, allowing for the quantitative and qualitative evaluation of the mutation burden in body fluids [87]. In this way, ctDNA has been considered as a good biomarker and can be utilized in disease monitoring. The data of ctDNA include quantitative changes, such as differences in the concentration of ctDNA, as well as qualitative changes, such as gene mutations, DNA copy number variations and DNA methylation [16]. Indicatively, studies based on the detection of somatic single-nucleotide mutations and methylation changes in ctDNA could closely correlate with tumor burden over time in HCC patients and could be used to predict recurrence after surgery [17,88,89]. Initially, it was shown that p15 and p16 methylations were positive in the plasma/serum of 92% of HCC patients [90]. In another study, Ras association domain family 1A(RASSF1A) promoter hypermethylation was detected in 90% of cases, with an overall predictive accuracy of 77.5%, compared to healthy controls [91]. In addition to RASSF1A, two abnormally methylated genes (APC and COX2) and one miRNA (miR-203) were combined to establish a predictive model by which nearly 75% of HCC patients were detected, who could not be diagnosed with AFP [92].

As ctDNA represents only a very small proportion of cell-free DNA, very sensitive and reliable detection methods are required. Levels of ctDNA are measured mainly by real-time PCR (RT-PCR) [93], while digital PCR (dPCR) [94] or sequencing methods are used for the detection of point mutations [95]. In addition to TERT and TP53 mutations as the prognostic factors of poor survival [45,47], other mutations have been shown to have prognostic values for HCC. MLH1 mutation was specifically associated with lower survival [1], whereas mutations of genes from the PI3K/mTOR pathway were shown to be the predictors of non-responders to TKI treatments for patients with advanced HCC [86].

A number of studies have also shown the prognostic values of circulating miRNAs in HCC. Lower survival rates were associated with patients with low levels of miR-1, miR-122, miR-26a, miR-29a and miR-223-3p [96,97,98,99] or high levels of miR-155, miR-96 and miR-193-5p [100,101]. Furthermore, six additional miRNAs were identified as prognostic factors. Low levels of miR-424-5p or miR- 101-3p and high levels of miR-128, miR-139-5p, miR-382-5p and miR410 were associated with lower survival rates in HCC patients [17]. Alternatively, miRNAs have been studied in association with EVs [32,33,34]. In a cohort of 59 HCC patients, a correlation was found between tumor recurrences after liver transplantation and a high level of exosomal miR-718 [102]. Additionally, high levels of exosomal miR-665 or low levels of exosomal miR-638 and miR-320a were identified as predictors of poor survival [103,104,105].

Another cornerstone of liquid biopsy is the isolation and detection of CTCs, which have been described as a useful tool for the prognostication of HCC [106]. As introduced above, EpCAM-positive CTC cells have been intensively investigated in HCC [50,51]. However, since CTCs can lose their epithelial phenotype through epithelial-to-mesenchymal transition (EMT) in order to survive and metastasize [49], EpCAM cannot always be considered an optimal biomarker to detect HCC. Alternatively, other phenotypic markers have been explored, such as the hepatocyte-specific asialoglycoprotein receptor (ASGPR) [67], and the hepatocyte paraffin 1 [107], or incorporation of several markers simultaneously, as it has extensively been reviewed elsewhere [20]. Most recently, in a prospective study of 80 HCC patients, a multimarker assay combining cell surface markers EpCAM, ASGPR and GPC3 was able to detect HCC CTCs in 97% of the patients with high accuracy. Moreover, a phenotypic variant subset of CTCs was associated with aggressive disease progression and underlying metastasis, therefore highlighting the important implications of CTCs in treatment selection [108]. Another study showed that the detection of phosphorylated ERK (pERK) and pAkt in CTCs could predict the response to sorafenib efficacy in advanced HCC patients, similarly to tumor tissue biopsy [59].

## 5. Clinical Applications for Immunotherapy in HCC

### 5.1. Immunotherapy in HCC

After years of sorafenib predominance and desolated perspectives, the skyline of systemic therapies for unresectable advanced HCC has considerably grown in the last decade. Not only more angiogenesis- and proliferation pathway-directed targeted therapies are available (TKIs; monoclonal antibodies), covering the therapeutic scenario from the first- to the third-line of treatment [109], but also ICIs are now well-established active agents and are gaining growing attention in the context of liver cancer [5]. Immuno-oncology (IO) represents a major breakthrough in this context, leading to a significant increase in median overall survival (OS) and to the possibility of long-term survival [110].

Nivolumab and pembrolizumab showed promising anti-tumoral activity with a 20% and 17% objective response rate (ORR), respectively, in patients who were refractory or intolerant to sorafenib in the phase 1/2 Checkmate-040 [6] and phase 2 Keynote-224 [111] clinical trials, respectively. Based on these results, the FDA granted an accelerated approval to the two anti-PD-1 antibodies for the second-line treatment of HCC. Notwithstanding these relevant and exciting results, the phase 3 trials testing nivolumab in the first line (Checkmate-459) [112] and pembrolizumab in the second line (Keynote-240) [113] of treatment failed to meet the protocol-defined statistical significance threshold of their primary endpoints. However, a clinically meaningful improvement in the overall response rate and complete responses were seen, confirming that ICIs are active and stressing their role for the treatment of liver cancer. Importantly, the safety profile was favorable and consistent with that of the primary analysis, supporting a favorable risk-to-benefit ratio. More recently, both the FDA and European Medicine Agency (EMA) approved the combination of atezolizumab (anti-PD-L1) and bevacizumab (anti-Vascular Endothelial Growth Factor-A monoclonal antibody) as a first-line systemic option for unresectable advanced-stage HCC, based on the striking results of the phase 3 trial IMbrave150 [114,115]. This study reported a 27% ORR and showed clinically meaningful and statistically significant improved outcomes for the experimental arm (vs. sorafenib) in terms of OS (NR (not reached) vs. 13.2 months, HR 0.58) and progression-free survival (6.8 vs. 4.3 months, HR 0.59) [4]. The treatment benefit was also meaningful after 12 months of additional follow-up, showing a median OS of 19.2 months with atezolizumab and bevacizumab vs. 13.4 months with sorafenib (HR 0.66, *p* = 0.0009), and a 29.8% ORR for the experimental arm, consistent with the primary analysis. Providing the longest survival ever reached in a first-line phase 3 trial in advanced HCC, this combination represents a practice-changing first-line treatment for HCC patients [116]. Following this stream, novel immuno-oncology-based combinations (ICI + ICI, ICI + TKI) are currently under development with the promise of improving the therapeutic management not only of advanced HCC but also of the early (in neo-adjuvant and adjuvant settings) and intermediate stages of the disease [110].

A common finding, that all trials, to date, have definitely revealed, is a significant heterogeneity in the magnitude of the tumor response to IO drugs and in the duration of the clinical response [117]. Despite a clear clinical benefit of immunotherapies in HCC, ICI showed efficacy only in a minority of HCC patients. A deeper knowledge of the dynamic interplay between all the components of the HCC ecosystem (tumor and immune cells, stromal cells, endothelial cells and nonmalignant cells) [1] is needed to dissect the mechanisms behind the clinical response to ICI and could result in an higher efficacy rate in the future.

Today, treatment allocation is not guided by any particular tumor characteristic, since no biomarker can effectively predict the response to a particular drug; the only exceptions, at present, are AFP and ramucirumab (anti-Vascular Endothelial Growth Factor Receptor-2 monoclonal antibody) [118,119]. Therefore, a crucial, urgent and still unmet need in the context of HCC research is the availability of robust and validated predictive biomarkers than can help in identifying the best candidates for a personalized IO approach [5,110,117,120]. Predictive biomarkers for the IO response and/or IO resistance might be extremely useful in order to optimize patient selection, spare toxicity to patients unlikely to respond and improve the design of clinical trials in the upcoming years [4,117].

The expression of PD-L1 on tumor cells, assessed by immunohistochemistry (IHC) from formalin-fixed paraffin-embedded (FFPE) tissue sections, is the only approved biomarker commonly used in the clinical routine to identify subgroups of patients with a higher chance of benefit from ICI [120,121]. This biomarker, however, is far from being perfect and does not confidently predict the response to cancer treatment. A few reasons can be indicated for this poor predictive value, for example, the discrepancy in PD-L1 IHC assessment in terms of the positivity cutoff, which is mostly related to the use of different detection antibodies or the choice of cell type to be stained [122]. Moreover, the unavailability of tissue or the low percentage of tumor cells in the biopsy makes this test not always feasible [120]. Despite the IHC measure, a proportion of PD-L1-“negative” patients will still respond and a proportion of PD-L1-“positive” patients will fail, showing how complex and still not fully unraveled the interplay is between cancer and the immune system [120,123]. In addition, tumor heterogeneity as a result of tumor evolution and subsequent clonal mutational differences [15] or due to treatment-induced resistant sub-clones [124] may further challenge disease monitoring and choice of therapy. These issues have prompted research to discover other markers, exploring not only tissue but also blood samples of cancer patients [121].

### 5.2. Liquid Biopsy for Immunotherapy in HCC

The race towards the identification of immunotherapy predictive biomarkers is at the forefront of research in HCC. Among the biomarkers of interest, there are TMB and mutational signatures identified from ctDNA, and PD-L1 expression detected on CTCs [125]. TMB and PD-L1 expression are considered good predictors in several cancers, but the evidence in liver cancer has not been as established thus far [117]. In Table 1, we summarize the most recent literature in the field, and we highlight the key findings for each study.

A proof-of-concept study was able to show that changes in the ctDNA levels could significantly correlate with tumor size in cancer patients treated with anti-PD1 drugs and be a valuable prognostic factor of progression-free and overall survival [126]. Another recent emerging predictor of immunotherapy efficacy is considered to be the TMB which is defined as the total number of somatic non-synonymous mutations per mega-base identified in tumor tissue or circulating tumor DNA [2,127]. In one study, targeted gene analysis of ctDNA showed high consistency in the levels of TMB between tissue and blood samples which were higher in TP53-mutated patients with advanced liver cancer, indicating that ctDNA analysis could be a better option to evaluate TMB prior to immunotherapy in cases of advanced primary liver cancers where tissue biopsy was not recommended [18]. However, in another prospective study of advanced HCC, mutational analysis of ctDNA could not be associated with response to ICI therapy but only to systemic treatment with TKIs [36], which implied that more and larger cohort clinical studies would be required in order to elucidate the potentials of mutational ctDNA analysis in determining immunotherapy efficacy.

In a small subset of patients enrolled in the phase 1b clinical trial receiving atezolizumab plus bevacizumab, higher levels of ctDNA at baseline were associated with an increased baseline tumor burden (*p* < 0.03), and ctDNA turned negative in 70%, 27%, 9% and 0% of patients achieving a complete response, partial response, stable disease and disease progression, respectively. Moreover, undetectable ctDNA levels during treatment were linked to a longer progression-free survival (PFS) [128], suggesting a role for the non-invasive ctDNA analysis in the prediction of immune response.

It has also been shown that a hyper-mutated ctDNA phenotype, the liquid counterpart of the tissue-based high level of TMB, was associated with a favorable outcome in a cohort of 69 cancer patients with different histologies, including three HCC patients, treated with different immune checkpoint inhibitors. In particular, the overall response rate, PFS and OS in the high-alteration groups, defined as variants of unknown significance (VUS) > 3 or total alterations ≥ 6, were significantly higher than in the low-alteration groups, defined as VUS ≤ 3 or total alterations < 6 (45% vs. 15% for high and low alterations, respectively, *p* = 0.014) [122].

Along with predictive biomarkers of response, the identification of predictive biomarkers of resistance is also compelling and might have relevant implications when designing future clinical trials. Among the oncogenic pathways that have been linked to potential tumor immunotherapy resistance, the Wnt/b-catenin signaling pathway is one of the earliest, with evidence coming from genomic, tissue and mouse model studies [129,130,131,132]. In this context, a preliminary study demonstrated that liquid biopsy was potentially able to detect Wnt/b-catenin-activating mutations in HCC [117]. However, another study could not confirm this hypothesis and postulated that the detection of Wnt/b-catenin pathway-activating mutations might not be sufficient to identify advanced HCC patients with primary resistance to ICI [119]. CTNNB1 is one of the genes involved in the Wnt/b-catenin pathway, and its mutations, such as p.T41A, are among the most prevalent genetic alterations in HCC [133]. Oversoe et al. evaluated the presence of the CTNNB1 p.T41A mutation comparing tumor tissue DNA and ctDNA and found that liquid biopsy managed to reveal mutations that were not detected in single tumor biopsies, thus increasing the detection rate of the CTNNB1 mutation in HCC patients and suggesting that ctDNA could empower the perspective of a tailored treatment strategy [134].

**Table 1 cancers-13-04334-t001:** Summary of the key findings in the most recent literature.

Type of Biomarker Analyzed	Key Findings	Reference
Changes in the ctDNA levels	Could significantly correlate with tumor size in cancer patients treated with anti-PD1 drugs and be a valuable prognostic factor of progression-free and overall survival.	[126]
Targeted gene analysis of ctDNA	Can be a better option to evaluate TMB prior to immunotherapy in cases of advanced primary liver cancers when tissue biopsy is not recommended.	[18]
Mutational analysis of ctDNA	Could not be associated with response to ICI therapy but only to systemic treatment.	[36]
Levels of ctDNA at baseline	Higher levels of ctDNA at baseline were associated with an increased baseline tumor burden, and ctDNA turned negative in 70%, 27%, 9% and 0% of patients achieving a complete response, partial response, stable disease and disease progression, respectively.	[128]
	Undetectable ctDNA levels during treatment were linked to a longer progression-free survival.	
Hyper-mutated ctDNA phenotype	Is associated with a favorable outcome in a cohort of 69 cancer patients with different histologies, including three HCC patients, treated with different immune checkpoint inhibitors.	[122]
	Overall response rate, PFS and OS in high-alteration groups were significantly higher than in low-alteration groups.	
Detection of Wnt/b-catenin-activating mutations	Wnt/b-catenin-activating mutations in HCC linked to potential tumor immunotherapy resistance in several studies.	[129,130,131,132]
Detection of Wnt/b-catenin-activating mutations	Demonstration that liquid biopsy is potentially able to detect Wnt/b-catenin-activating mutations in HCC.	[117]
Detection of Wnt/b-catenin-activating mutations	Detection of Wnt/b-catenin pathway-activating mutations might not be sufficient to identify advanced HCC patients with primary resistance to ICI.	[119]
Targeted mutational analysis of CTNNB1 p.T41A mutation	ctDNA liquid biopsy managed to reveal mutations that were not detected in single tumor biopsies, thus increasing the detection rate of CTNNB1 mutation in HCC patients.	[134]
PD-L1 expression on CTCs	Biomarker to assess ICI-based immunotherapy efficacy of advanced solid tumors.	[135]
CTCs expressing PD-L1	PD-L1-positive CTCs are mainly found in advanced stages of disease, and they represent an independent prognostic factor for overall survival. 6 out of 10 patients receiving anti-PD-1 treatment had PD-L1-positive CTCs at baseline, and of these, 5 out of 6 had a favorable treatment response. 4 out of 10 patients receiving anti-PD-1 treatment did not have PD-L1+ CTCs and were non-responders.	[136]

Apart from mutational signatures and TMB in connection to ctDNA, CTCs expressing PD-L1 have been suggested as another promising biomarker to assess the ICI-based immunotherapy efficacy of advanced solid tumors [135]. Winograd and colleagues analyzed the expression of PD-L1 on CTCs and found that PD-L1-positive CTCs are mainly found in the advanced stages of disease and that they represent an independent prognostic factor for overal survival. Moreover, of 10 patients receiving anti-PD-1 treatment, 6 had PD-L1-positive CTCs at baseline, and of these, 5 had a favorable treatment response, whereas 4 patients did not have PD-L1+ CTCs and were non-responders [136]. They definitely unraveled the prognostic and potentially predictive value of CTCs in the context of HCC. Clearly, as pointed out by Hofman and coauthors, together with technical issues, some relevant questions could also be asked: e.g., (1) Is PD-L1 expression in CTCs correlated with matched tissue samples? (2) Is this expression homogeneous on all CTCs or is it restricted to a subpopulation of CTCs? (3) What is the implication of PD-L1 expression in circulating immune cells when associated with CTCs? [120]

Undoubtedly, liquid biopsy-based biomarkers for immunotherapy in HCC require further rigorous testing and validation in carefully and well-designed clinical trials where their performance can be evaluated and their clinical implication can be measured. Hopefully, liquid biopsy will be integrated in the daily clinical management of HCC in the near future.

Notably, it is likely that immune-genomic biomarkers may further enhance immunotherapy through novel additional checkpoint inhibitors, but also neoantigen vaccine or adoptive cell transfer approaches [5,137]. This can certainly be of paramount importance to magnify the immunotherapy treatment response and implement precision immune-oncology.

## 6. Limitations and Future Perspectives

Immunotherapy represents the latest and most promising clinical development for the treatment of advanced HCC. However, despite the great benefits immunotherapies have brought, there is still a significant number of cancer patients who do not respond or respond poorly to these new approaches [6]. Therefore, there is an imperative need to increase the reliability and enrich the repertoire of currently available biomarkers, in order to more accurately predict therapy efficacy and to improve response rates, compared to existing therapies. In particular, the attractive types of biomarkers are the ones that can be monitored without invasive procedures. Liquid biopsies, from which it is possible to isolate ctDNA and CTCs circulating in the blood stream of cancer patients, have shown promising data as prognostic and diagnostic tools for HCC while, at the same time, also allowing sequential and real-time monitoring of disease status in a minimally invasive manner [16,22,138]. This is especially important for patients with advanced and unresectable HCC, when surgical and other invasive procedures are not an option [18]. Interestingly, in some cases, ctDNA has been shown to be superior in identifying mutational signatures that could not be traced in single tumor biopsies [134] or that could correlate more closely with the tumor load and predict treatment efficacy with higher sensitivity, compared to AFP or imaging in patients with unresectable liver cancer [18]. These data prove the predictive value of analyzing ctDNA, which can add to the existing and well-established diagnostic tools for HCC. However, other studies have not been able to confirm the association of tumoral signature mutations with resistance to ICIs in HCC based on their ctDNA analysis [36]. These conflicting data highlight that larger and more comprehensive clinical studies are required, in order to obtain widely applicable and consistent results, which, at the moment, are missing in the field of liquid biopsy for HCC. CTCs are another important source of biomarkers, and many studies point out their role in the prognosis of HCC [51]. Importantly, CTCs have also been used to prognosticate responses to ICIs through the expression of PD-L1 in patients with advanced liver cancer [136]. However, low levels of CTCs in the early stages of HCC and the lack of standardized procedures make it challenging to integrate CTC techniques in the clinical practice for HCC diagnosis [51].

In this context, factors associated with the individual’s genetic background, the tumor microenvironment and interactions with the host immune system may additionally challenge the selection and evaluation of biomarkers able to predict tumor responses. In order to overcome these issues, integration of multiple biomarkers rather than single analytes as well as combinational approaches based on genomic and proteomic analyses will most probably be able to improve the precision of personalized treatments. Here, the implementation of novel NGS technologies and artificial intelligence might be of great importance to identify genomic and immunologic signatures predictive of treatment responses, as it has encouragingly been shown in recent studies [73,139,140,141].

Up to the present, ICIs have been the most widely used immune-based approaches in the clinical management of advanced HCC [110]. Therefore, the majority of cancer immunotherapy predictive biomarkers described thus far refer mainly to responses to ICIs. Other immunotherapy modalities such as CAR-T cell therapies and neoantigen vaccines are currently being tested in ongoing clinical studies for HCC, and the preliminary results are awaited with great interest [2,5].

## 7. Conclusions

Although liquid biopsy biomarkers for HCC are not well evaluated, compared to other malignancies, there are many studies highlighting their prognostic and diagnostic value for clinical management. Here, we focus on ctDNA and CTCs, two emerging liquid biopsy biomarkers, and their role in HCC immunotherapy. In the near future, the identification and further validation of novel and existing biomarkers, as well as incorporation of high-throughput technologies, will remain of paramount importance in order to improve precision in treatment decision making in the field of immuno-oncology, especially for HCC immunotherapy.

## Figures and Tables

**Figure 1 cancers-13-04334-f001:**
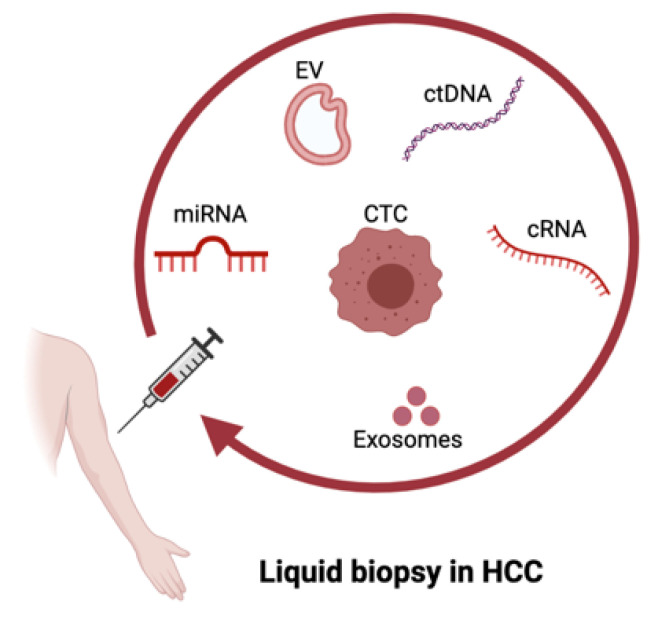
Liquid biopsy in hepatocellular carcinoma (HCC). Illustration of liquid biopsy biomarkers investigated in HCC, including circulating nucleic acids, circulating tumor DNA (ctDNA), circulating RNA (cRNA)/microRNAs (miRNA), extracellular vesicles (EVs)/exosomes and circulating tumor cells (CTC). Created with BioRender.com accessed on 11 August 2021.

**Figure 2 cancers-13-04334-f002:**
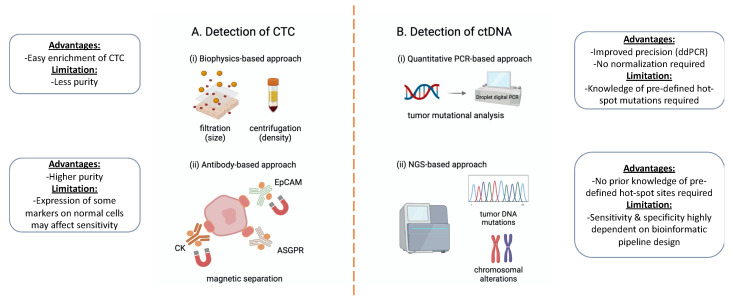
Main technology platforms utilized for the detection of CTCs and ctDNA in HCC, together with advantages and limitations for each approach. (**A**) The detection of CTCs has been based either on biophysics-based properties, including size for filtration and density by centrifugation (**i**), or antibody-based approaches targeting known CTC expression markers, such as the epithelial cell adhesion molecule (EpCAM), cytokeratin markers (CK) and asialoglycoprotein receptor (ASGPR). Antibodies enriched with magnetic beads can be subsequently magnetically separated to isolate CTCs (**ii**). (**B**) For the detection of ctDNA, various quantitative PCR-based approaches have been utilized, such as droplet digital PCR (ddPCR), to improve the precision of mutational analysis of ctDNA (**i**). The next-generation sequencing (NGS)-based approach represents another alternative platform for ctDNA detection. A unique advantage of NGS is that it can detect tumor DNA mutations in plasma without prior knowledge of pre-defined hotspot mutation sites. Apart from DNA mutations, chromosomal alterations can also be detected by this approach (**ii**). Created with BioRender.com accessed on 11 August 2021.

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
