# Peer review of "Liquid Biopsy in Hepatocellular Carcinoma: Opportunities and Challenges for Immunotherapy"

_cancers, 2021, doi:10.3390/cancers13174334_

Round 1

Reviewer 1 Report

General

In this review article entitled “Liquid biopsy in hepatocellular carcinoma: opportunity and challenges for immunotherapy”, the authors mentioned that there were no effective predictive markers for response to immunotherapy or for recurrence after surgical therapy. To solve the problem, they proposed the clinical application of liquid biopsies with circulating tumor cells and circulating tumor DNA for immunotherapy of HCC. This review is interesting. However, there are some concerns to be clarified. Some areas of particular concern are outlined below.  

Major points

  • In “Liquid biopsy as a diagnostic and prognostic tool”, the authors showed the utility of ctDNA and CTCs as a diagnostic and prognostic tools of HCC, they also discuss the possibility of temporal evaluation of ctDNA and CTCs for early detection of HCC recurrence.

Minor points

  • Page 2, line 5; “Sorafenib and Lenvatinib” as well as Nivolumab, Pembrolizumab, Atezolizumab and Bevacizumab are usually expressed as a small letter.
  • Page 8, line 42; “in” is duplicated.
  • Page 9, line 6; The up-dated OS of atezolizumab and bevacizumab is 19.2 months. The authors should referred the latest data of IMbrave 150.

Author Response

Response to Reviewer 1

General

In this review article entitled “Liquid biopsy in hepatocellular carcinoma: opportunity and challenges for immunotherapy”, the authors mentioned that there were no effective predictive markers for response to immunotherapy or for recurrence after surgical therapy. To solve the problem, they proposed the clinical application of liquid biopsies with circulating tumor cells and circulating tumor DNA for immunotherapy of HCC. This review is interesting. However, there are some concerns to be clarified. Some areas of particular concern are outlined below.  

We thank the reviewer for this summary that captures the intended main message of our review. We are happy to address the concerns in the following sections.

Major points

  • In “Liquid biopsy as a diagnostic and prognostic tool”, the authors showed the utility of ctDNA and CTCs as a diagnostic and prognostic tools of HCC, they also discuss the possibility of temporal evaluation of ctDNA and CTCs for early detection of HCC recurrence.

In this session we wanted to summarize the recent applications of liquid biopsy as diagnostic and prognostic tool for HCC, but we also had text referring to early detection of HCC recurrence, that was probably better suited for a different section as the reviewer pointed out. To give a better flow to the review we move the text “Detection of EpCAM positive CTCs with co-existing T regulatory cells (CD4+/CD25+/Foxp3+) indicated HCC recurrence [52]. In this study patients with high CTC/Tregs levels had a significantly higher risk of developing postoperative HCC recurrence than those with low CTC and Treg levels. In addition, other subtypes of CTCs have been explored, including mesenchymal CTCs the presence of which has been also associated with higher risk of tumor recurrence in HCC patients [53, 54].” To section 2.2 “CTCs in HCC”.

In section 4 we also edited all the text indicated in yellow by the editor to eliminate repetitions and improve the readability.

Minor points

  • Page 2, line 5; “Sorafenib and Lenvatinib” as well as Nivolumab, Pembrolizumab, Atezolizumab and Bevacizumab are usually expressed as a small letter.

We corrected the nomenclature accordingly (underlined in red in main text).

  • Page 8, line 42; “in” is duplicated.

We thank the reviewer for noticing this repetition that we have now eliminated.

  • Page 9, line 6; The up-dated OS of atezolizumab and bevacizumab is 19.2 months. The authors should referred the latest data of IMbrave 150.

We thank the reviewer for this important update, and we have now corrected accordingly in main text as following: “Treatment benefit was meaningful also after 12 months of additional follow up, showing a median OS of 19.2 months with atezolizumab and bevacizumab vs 13.4 months with sorafenib (HR 0.66, p=0.0009) and a 29.8% ORR for the experimental arm, consistent with the primary analysis. Providing the longest survival ever reached in a first-line phase 3 trial in advanced HCC, this combination represents a practice-changing first-line treatment for HCC patients.”. We have also included a new reference: Richard S. Finn, Shukui Qin, Masafumi Ikeda, Peter R. Galle, Michel Ducreux, Tae-You Kim, Ho Yeong Lim, Masatoshi Kudo, Valeriy Vladimirovich Breder, Philippe Merle, Ahmed Omar Kaseb, Daneng Li, Wendy Verret, Hui Shao, Juan Liu, Lindong Li, Andrew X. Zhu, and Ann-Lii Cheng. Updated overall survival (OS) data from a global, randomized, open-label phase III study of atezolizumab (atezo) + bevacizumab (bev) versus sorafenib (sor) in patients (pts) with unresectable hepatocellular carcinoma (HCC). Journal of Clinical Oncology, 2021. 39:3_suppl, 267-267

Reviewer 2 Report

The manuscript entitled:" Liquid biopsy in hepatocellular carcinoma: opportunity and challenges for immunotherapy" focused on a systemic analysis of literature data about liquid biopsy as promising tool in the clinical administration of HCC patients represents a fascinating paper where the most relevant aspects of plasma derived biomarkers are discussed by the authors. In my opinion, few minor comments should be implemented to accept this paper for the publication.

  • In the introduction section (lines 58- 72) the authors highlighted the current biomarkers available for HCC patients tested o ntissue samples. In my opinion, the authors could improve this section with a brief description of actually tested tissue based biomarkers yet approved in clinical practice or invetigated in ongoing clinical trials. In addition, they also stress the limitation of tissue samples, in particular in some clinical setting.

- In the text, the authors clearly define the main clinical setting where this approach may help to decision making about HCC patients in clinical setting. In my opinion, the authors could identify some relevant papers where liquid biopsy is adopted as monitoring tool for HCC patients. This aspect allows to clinically imporve the quality of the manuscript

Author Response

Response to Reviewer 2

The manuscript entitled:" Liquid biopsy in hepatocellular carcinoma: opportunity and challenges for immunotherapy" focused on a systemic analysis of literature data about liquid biopsy as promising tool in the clinical administration of HCC patients represents a fascinating paper where the most relevant aspects of plasma derived biomarkers are discussed by the authors. In my opinion, few minor comments should be implemented to accept this paper for the publication.

We thank the reviewer for this summary, and we are happy to address each concern in the following sections.

  • In the introduction section (lines 58- 72) the authors highlighted the current biomarkers available for HCC patients tested o ntissue samples. In my opinion, the authors could improve this section with a brief description of actually tested tissue based biomarkers yet approved in clinical practice or invetigated in ongoing clinical trials. In addition, they also stress the limitation of tissue samples, in particular in some clinical setting.

We added the text “Higher TMB based on genomic profiling of various tumor biopsies could reflect higher likelihood for response to ICIs [9] while with regards to PD-L1 testing, it could positively correlate with better responses to anti-PD-L1 therapy [10, 11]. In addition, gene expression analysis on HCC adjacent tumor tissues has been able to identify signatures correlated with improved survival [12].” In the introduction.

We also added the text “In addition, unlikely to other solid tumors, tissue biopsies for HCC are not frequently available since diagnosis mainly relies on imaging [17]. Also at a late stage, when the lesions are unresectable, liver biopsy is usually not recommended for advanced HCC [18] while there is risk of extrahepatic tumor spread along the needle track [19].

- In the text, the authors clearly define the main clinical setting where this approach may help to decision making about HCC patients in clinical setting. In my opinion, the authors could identify some relevant papers where liquid biopsy is adopted as monitoring tool for HCC patients. This aspect allows to clinically imporve the quality of the manuscript

We have added references to:

  • Alunni-Fabbroni, M., K. Rönsch, T. Huber, C.C. Cyran, M. Seidensticker, J. Mayerle, M. Pech, B. Basu, C. Verslype, J. Benckert, et al., Circulating DNA as prognostic biomarker in patients with advanced hepatocellular carcinoma: a translational exploratory study from the SORAMIC trial. J Transl Med, 2019. 17(1): p. 328.
  • Huang, A., X. Zhang, S.L. Zhou, Y. Cao, X.W. Huang, J. Fan, X.R. Yang, and J. Zhou, Plasma Circulating Cell-free DNA Integrity as a Promising Biomarker for Diagnosis and Surveillance in Patients with Hepatocellular Carcinoma. J Cancer, 2016. 7(13): p. 1798-1803.
  • Xiong, Y., C.R. Xie, S. Zhang, J. Chen, and Z.Y. Yin, Detection of a novel panel of somatic mutations in plasma cell-free DNA and its diagnostic value in hepatocellular carcinoma. Cancer Manag Res, 2019. 11: p. 5745-5756.
  • El-Tawdi, A.H., M. Matboli, H.H. Shehata, F. Tash, N. El-Khazragy, S. Azazy Ael, and O. Abdel-Rahman, Evaluation of Circulatory RNA-Based Biomarker Panel in Hepatocellular Carcinoma. Mol Diagn Ther, 2016. 20(3): p. 265-77.
  • Moshiri, F., A. Salvi, L. Gramantieri, A. Sangiovanni, P. Guerriero, G. De Petro, C. Bassi, L. Lupini, A. Sattari, D. Cheung, et al., Circulating miR-106b-3p, miR-101-3p and miR-1246 as diagnostic biomarkers of hepatocellular carcinoma. Oncotarget, 2018. 9(20): p. 15350-15364.
  • Yamamoto, Y., S. Kondo, J. Matsuzaki, M. Esaki, T. Okusaka, K. Shimada, Y. Murakami, M. Enomoto, A. Tamori, K. Kato, et al., Highly Sensitive Circulating MicroRNA Panel for Accurate Detection of Hepatocellular Carcinoma in Patients With Liver Disease. Hepatol Commun, 2020. 4(2): p. 284-297.
  • Kelley, R.K., M.J. Magbanua, T.M. Butler, E.A. Collisson, J. Hwang, N. Sidiropoulos, K. Evason, R.M. McWhirter, B. Hameed, E.M. Wayne, et al., Circulating tumor cells in hepatocellular carcinoma: a pilot study of detection, enumeration, and next-generation sequencing in cases and controls. BMC Cancer, 2015. 15: p. 206.
  • von Felden, J., K. Schulze, T. Krech, F. Ewald, B. Nashan, K. Pantel, A.W. Lohse, S. Riethdorf, and H. Wege, Circulating tumor cells as liquid biomarker for high HCC recurrence risk after curative liver resection. Oncotarget, 2017. 8(52): p. 89978-89987.
  • Julich-Haertel, H., S.K. Urban, M. Krawczyk, A. Willms, K. Jankowski, W. Patkowski, B. Kruk, M. KrasnodÄ™bski, J. Ligocka, R. Schwab, et al., Cancer-associated circulating large extracellular vesicles in cholangiocarcinoma and hepatocellular carcinoma. J Hepatol, 2017. 67(2): p. 282-292.
  • Wang, Y., C. Zhang, P. Zhang, G. Guo, T. Jiang, X. Zhao, J. Jiang, X. Huang, H. Tong, and Y. Tian, Serum exosomal microRNAs combined with alpha-fetoprotein as diagnostic markers of hepatocellular carcinoma. Cancer Med, 2018. 7(5): p. 1670-1679.
  • Tucci, M., A. Passarelli, F. Mannavola, L.S. Stucci, P.A. Ascierto, M. Capone, G. Madonna, P. Lopalco, and F. Silvestris, Serum exosomes as predictors of clinical response to ipilimumab in metastatic melanoma. Oncoimmunology, 2018. 7(2): p. e1387706.

Reviewer 3 Report

The article by Panagiota Maravelia et al. entitled “Liquid biopsy in hepatocellular carcinoma: opportunity and challenges for immunotherapy” is announced as a review about use of liquid biopsies in the context of hepatocellular carcinoma but it is a bit weak. The authors have to add much more information to this actual and hot biomedical theme and they have to improve the manuscript before it can be published in the journal “Cancers”.

My major concerns (in addition to the one already mentioned above) are:

- In lines 34-35 all molecules presented in Figure 1 must be included.

- When the authors write “A number of clinical trials….”, “” or similar more than a single citation is necessary.

- Very often citations for statements are missing; e.g. lines 70, 74, 75, 77, 81, 98, 101, 103, 131. 132, 134, 135, 137, 138, 143, 156, 157, 164, 167, 171, 173, 182, 185, 187, 189, 197, 200, 201, 210, 212, 213, 218, 220, 223, 226, 229, 230, 231, 234, 236, 238, 240, 241, 242, 245, 285, 290, 294, 298, 314, 316, 324, 330, 336, 346, 347, 358, 364, 365, 437, 443, 470, 474, 495, 501, 504. 505, 507, 512, 526, 529, 536 (of note this list is not complete).

- Why wrote the authors “….extracellular vesicles and exosomes ….”? Per definition exosomes are a class of extracellular vesicles and must not be mentioned separately.

- All used abbreviations in Figure 1 must be explained in the Figure legend.

- The authors have to introduce all abbreviations before using these; e.g. PBMCs. Of course, an abbreviation should be only introduced if it is used later in the text.

- It is neither nice nor necessary to read “ ….by Zhao et al ….” or “…Labgaa et al..” etc. if somebody is interested in the name of the first author (s)he will find this information in the Reference list. The authors have to rephrase all these parts.

- The authors wrote “..other phenotypic markers have been explored using different technology platforms and assays…” This is a bit too superficial and here the different technology platforms and assays must be added.

- Also the statement “This group has….” must be rephrased.

- The statement “…., etc.” is not scientific and the authors have to mention here all genes. The same must be done in line 284.

- The authors have to mention in detail the mutations and should not simply write “..other mutations…” On top the authors have not added any citations.

- Under the point 3.2 one of the biggest advantages of ddPCR is missing – it is possible to measure a target without normalization to a housekeeping.

- Under point 5 the authors have to add the aspect of tumour heterogeneity.

Author Response

Response to Reviewer 3

The article by Panagiota Maravelia et al. entitled “Liquid biopsy in hepatocellular carcinoma: opportunity and challenges for immunotherapy” is announced as a review about use of liquid biopsies in the context of hepatocellular carcinoma but it is a bit weak. The authors have to add much more information to this actual and hot biomedical theme and they have to improve the manuscript before it can be published in the journal “Cancers”.

We thank the reviewer for the suggestions and critical revision that helped us to greatly improve the manuscript.

My major concerns (in addition to the one already mentioned above) are:

- In lines 34-35 all molecules presented in Figure 1 must be included.

We have now accordingly included all molecules presented in Figure 1 in the abstract session (underlined in red in main text).

- When the authors write “A number of clinical trials….”, “” or similar more than a single citation is necessary.

We added the following references:

  • Lai, E., G. Astara, P. Ziranu, A. Pretta, M. Migliari, M. Dubois, C. Donisi, S. Mariani, N. Liscia, V. Impera, et al., Introducing immunotherapy for advanced hepatocellular carcinoma patients: Too early or too fast? Crit Rev Oncol Hematol, 2021. 157: p. 103167.
  • Sangro, B., P. Sarobe, S. Hervás-Stubbs, and I. Melero, Advances in immunotherapy for hepatocellular carcinoma. Nat Rev Gastroenterol Hepatol, 2021: p. 1-19.
  • Köberle, V., B. Kronenberger, T. Pleli, J. Trojan, E. Imelmann, J. Peveling-Oberhag, M.W. Welker, M. Elhendawy, S. Zeuzem, A. Piiper, et al., Serum microRNA-1 and microRNA-122 are prognostic markers in patients with hepatocellular carcinoma. Eur J Cancer, 2013. 49(16): p. 3442-9.
  • Xu, Y., X. Bu, C. Dai, and C. Shang, High serum microRNA-122 level is independently associated with higher overall survival rate in hepatocellular carcinoma patients. Tumour Biol, 2015. 36(6): p. 4773-6.
  • Cho, H.J., S.S. Kim, J.S. Nam, J.K. Kim, J.H. Lee, B. Kim, H.J. Wang, B.W. Kim, J.D. Lee, D.Y. Kang, et al., Low levels of circulating microRNA-26a/29a as poor prognostic markers in patients with hepatocellular carcinoma who underwent curative treatment. Clin Res Hepatol Gastroenterol, 2017. 41(2): p. 181-189.
  • Pratedrat, P., N. Chuaypen, P. Nimsamer, S. Payungporn, N. Pinjaroen, B. Sirichindakul, and P. Tangkijvanich, Diagnostic and prognostic roles of circulating miRNA-223-3p in hepatitis B virus-related hepatocellular carcinoma. PLoS One, 2020. 15(4): p. e0232211.

- Very often citations for statements are missing; e.g. lines 70, 74, 75, 77, 81, 98, 101, 103, 131. 132, 134, 135, 137, 138, 143, 156, 157, 164, 167, 171, 173, 182, 185, 187, 189, 197, 200, 201, 210, 212, 213, 218, 220, 223, 226, 229, 230, 231, 234, 236, 238, 240, 241, 242, 245, 285, 290, 294, 298, 314, 316, 324, 330, 336, 346, 347, 358, 364, 365, 437, 443, 470, 474, 495, 501, 504. 505, 507, 512, 526, 529, 536 (of note this list is not complete).

We thank the reviewer for this very appropriate comment, and we have added 29 new citations as listed below:

  • Alunni-Fabbroni, M., K. Rönsch, T. Huber, C.C. Cyran, M. Seidensticker, J. Mayerle, M. Pech, B. Basu, C. Verslype, J. Benckert, et al., Circulating DNA as prognostic biomarker in patients with advanced hepatocellular carcinoma: a translational exploratory study from the SORAMIC trial. J Transl Med, 2019. 17(1): p. 328.
  • Huang, A., X. Zhang, S.L. Zhou, Y. Cao, X.W. Huang, J. Fan, X.R. Yang, and J. Zhou, Plasma Circulating Cell-free DNA Integrity as a Promising Biomarker for Diagnosis and Surveillance in Patients with Hepatocellular Carcinoma. J Cancer, 2016. 7(13): p. 1798-1803.
  • Xiong, Y., C.R. Xie, S. Zhang, J. Chen, and Z.Y. Yin, Detection of a novel panel of somatic mutations in plasma cell-free DNA and its diagnostic value in hepatocellular carcinoma. Cancer Manag Res, 2019. 11: p. 5745-5756.
  • El-Tawdi, A.H., M. Matboli, H.H. Shehata, F. Tash, N. El-Khazragy, S. Azazy Ael, and O. Abdel-Rahman, Evaluation of Circulatory RNA-Based Biomarker Panel in Hepatocellular Carcinoma. Mol Diagn Ther, 2016. 20(3): p. 265-77.
  • Moshiri, F., A. Salvi, L. Gramantieri, A. Sangiovanni, P. Guerriero, G. De Petro, C. Bassi, L. Lupini, A. Sattari, D. Cheung, et al., Circulating miR-106b-3p, miR-101-3p and miR-1246 as diagnostic biomarkers of hepatocellular carcinoma. Oncotarget, 2018. 9(20): p. 15350-15364.
  • Yamamoto, Y., S. Kondo, J. Matsuzaki, M. Esaki, T. Okusaka, K. Shimada, Y. Murakami, M. Enomoto, A. Tamori, K. Kato, et al., Highly Sensitive Circulating MicroRNA Panel for Accurate Detection of Hepatocellular Carcinoma in Patients With Liver Disease. Hepatol Commun, 2020. 4(2): p. 284-297.
  • Kelley, R.K., M.J. Magbanua, T.M. Butler, E.A. Collisson, J. Hwang, N. Sidiropoulos, K. Evason, R.M. McWhirter, B. Hameed, E.M. Wayne, et al., Circulating tumor cells in hepatocellular carcinoma: a pilot study of detection, enumeration, and next-generation sequencing in cases and controls. BMC Cancer, 2015. 15: p. 206.
  • Julich-Haertel, H., S.K. Urban, M. Krawczyk, A. Willms, K. Jankowski, W. Patkowski, B. Kruk, M. KrasnodÄ™bski, J. Ligocka, R. Schwab, et al., Cancer-associated circulating large extracellular vesicles in cholangiocarcinoma and hepatocellular carcinoma. J Hepatol, 2017. 67(2): p. 282-292.
  • Wang, Y., C. Zhang, P. Zhang, G. Guo, T. Jiang, X. Zhao, J. Jiang, X. Huang, H. Tong, and Y. Tian, Serum exosomal microRNAs combined with alpha-fetoprotein as diagnostic markers of hepatocellular carcinoma. Cancer Med, 2018. 7(5): p. 1670-1679.
  • Tucci, M., A. Passarelli, F. Mannavola, L.S. Stucci, P.A. Ascierto, M. Capone, G. Madonna, P. Lopalco, and F. Silvestris, Serum exosomes as predictors of clinical response to ipilimumab in metastatic melanoma. Oncoimmunology, 2018. 7(2): p. e1387706.
  • Ryder, S.D., Guidelines for the diagnosis and treatment of hepatocellular carcinoma (HCC) in adults. Gut, 2003. 52 Suppl 3(Suppl 3): p. iii1-8.
  • Silva, M.A., B. Hegab, C. Hyde, B. Guo, J.A. Buckels, and D.F. Mirza, Needle track seeding following biopsy of liver lesions in the diagnosis of hepatocellular cancer: a systematic review and meta-analysis. Gut, 2008. 57(11): p. 1592-6.
  • Plaks, V., C.D. Koopman, and Z. Werb, Cancer. Circulating tumor cells. Science, 2013. 341(6151): p. 1186-8.
  • Mao, X., C. Liu, H. Tong, Y. Chen, and K. Liu, Principles of digital PCR and its applications in current obstetrical and gynecological diseases. Am J Transl Res, 2019. 11(12): p. 7209-7222.
  • Jamal-Hanjani, M., S.A. Quezada, J. Larkin, and C. Swanton, Translational implications of tumor heterogeneity. Clin Cancer Res, 2015. 21(6): p. 1258-66.
  • McGranahan, N. and C. Swanton, Biological and therapeutic impact of intratumor heterogeneity in cancer evolution. Cancer Cell, 2015. 27(1): p. 15-26.
  • Davis, A.A. and V.G. Patel, The role of PD-L1 expression as a predictive biomarker: an analysis of all US Food and Drug Administration (FDA) approvals of immune checkpoint inhibitors. J Immunother Cancer, 2019. 7(1): p. 278.
  • Goodman, A.M., S. Kato, L. Bazhenova, S.P. Patel, G.M. Frampton, V. Miller, P.J. Stephens, G.A. Daniels, and R. Kurzrock, Tumor Mutational Burden as an Independent Predictor of Response to Immunotherapy in Diverse Cancers. Mol Cancer Ther, 2017. 16(11): p. 2598-2608.
  • Kowanetz, M., W. Zou, S.N. Gettinger, H. Koeppen, M. Kockx, P. Schmid, E.E. Kadel, 3rd, I. Wistuba, J. Chaft, N.A. Rizvi, et al., Differential regulation of PD-L1 expression by immune and tumor cells in NSCLC and the response to treatment with atezolizumab (anti-PD-L1). Proc Natl Acad Sci U S A, 2018. 115(43): p. E10119-e10126.
  • Herbst, R.S., P. Baas, D.W. Kim, E. Felip, J.L. Pérez-Gracia, J.Y. Han, J. Molina, J.H. Kim, C.D. Arvis, M.J. Ahn, et al., Pembrolizumab versus docetaxel for previously treated, PD-L1-positive, advanced non-small-cell lung cancer (KEYNOTE-010): a randomised controlled trial. Lancet, 2016. 387(10027): p. 1540-1550.
  • Hoshida, Y., A. Villanueva, M. Kobayashi, J. Peix, D.Y. Chiang, A. Camargo, S. Gupta, J. Moore, M.J. Wrobel, J. Lerner, et al., Gene expression in fixed tissues and outcome in hepatocellular carcinoma. N Engl J Med, 2008. 359(19): p. 1995-2004.
  • Quandt, D., H. Dieter Zucht, A. Amann, A. Wulf-Goldenberg, C. Borrebaeck, M. Cannarile, D. Lambrechts, H. Oberacher, J. Garrett, T. Nayak, et al., Implementing liquid biopsies into clinical decision making for cancer immunotherapy. Oncotarget, 2017. 8(29): p. 48507-48520.
  • Cohen, J.D., L. Li, Y. Wang, C. Thoburn, B. Afsari, L. Danilova, C. Douville, A.A. Javed, F. Wong, A. Mattox, et al., Detection and localization of surgically resectable cancers with a multi-analyte blood test. Science, 2018. 359(6378): p. 926-930.
  • Chakrabarti, S., H. Xie, R. Urrutia, and A. Mahipal, The Promise of Circulating Tumor DNA (ctDNA) in the Management of Early-Stage Colon Cancer: A Critical Review. Cancers (Basel), 2020. 12(10).
  • Cai, J., L. Chen, Z. Zhang, X. Zhang, X. Lu, W. Liu, G. Shi, Y. Ge, P. Gao, Y. Yang, et al., Genome-wide mapping of 5-hydroxymethylcytosines in circulating cell-free DNA as a non-invasive approach for early detection of hepatocellular carcinoma. Gut, 2019. 68(12): p. 2195-2205.
  • Kaseb, A.O., N.S. Sánchez, S. Sen, R.K. Kelley, B. Tan, A.G. Bocobo, K.H. Lim, R. Abdel-Wahab, M. Uemura, R.C. Pestana, et al., Molecular Profiling of Hepatocellular Carcinoma Using Circulating Cell-Free DNA. Clin Cancer Res, 2019. 25(20): p. 6107-6118.
  • Tran, N.H., J. Kisiel, and L.R. Roberts, Using cell-free DNA for HCC surveillance and prognosis. JHEP Rep, 2021. 3(4): p. 100304.
  • Lamps, L.W. and A.L. Folpe, The diagnostic value of hepatocyte paraffin antibody 1 in differentiating hepatocellular neoplasms from nonhepatic tumors: a review. Adv Anat Pathol, 2003. 10(1): p. 39-43.
  • Finn, R.S., S. Qin, M. Ikeda, P.R. Galle, M. Ducreux, T.-Y. Kim, H.Y. Lim, M. Kudo, V.V. Breder, P. Merle, et al., IMbrave150: Updated overall survival (OS) data from a global, randomized, open-label phase III study of atezolizumab (atezo) + bevacizumab (bev) versus sorafenib (sor) in patients (pts) with unresectable hepatocellular carcinoma (HCC). Journal of Clinical Oncology, 2021. 39(3_suppl): p. 267-267.

In addition to the newly included citations listed above, already existing references have been cited in main text where missing before which can be identified through the ‘track changes’ mode.

- Why wrote the authors “….extracellular vesicles and exosomes ….”? Per definition exosomes are a class of extracellular vesicles and must not be mentioned separately.

We agree with the reviewer about this discrepancy, we used this wording referring to “Lee, E.Y. and R.P. Kulkarni, Circulating biomarkers predictive of tumor response to cancer immunotherapy. Expert Rev Mol Diagn, 2019. 19(10): p. 895-904”, where exosomes are referred separately from extracellular vesicles, perhaps in order to highlight their potentials as biomarker in cancer immunotherapy. However, we rephrased to “extracellular vesicles (e.g exosomes)” including respective citations:

  • Julich-Haertel, H., S.K. Urban, M. Krawczyk, A. Willms, K. Jankowski, W. Patkowski, B. Kruk, M. KrasnodÄ™bski, J. Ligocka, R. Schwab, et al., Cancer-associated circulating large extracellular vesicles in cholangiocarcinoma and hepatocellular carcinoma. J Hepatol, 2017. 67(2): p. 282-292.
  • Wang, Y., C. Zhang, P. Zhang, G. Guo, T. Jiang, X. Zhao, J. Jiang, X. Huang, H. Tong, and Y. Tian, Serum exosomal microRNAs combined with alpha-fetoprotein as diagnostic markers of hepatocellular carcinoma. Cancer Med, 2018. 7(5): p. 1670-1679.
  • Tucci, M., A. Passarelli, F. Mannavola, L.S. Stucci, P.A. Ascierto, M. Capone, G. Madonna, P. Lopalco, and F. Silvestris, Serum exosomes as predictors of clinical response to ipilimumab in metastatic melanoma. Oncoimmunology, 2018. 7(2): p. e1387706.

- All used abbreviations in Figure 1 must be explained in the Figure legend.

We have now added explanations for all abbreviations.

- The authors have to introduce all abbreviations before using these; e.g. PBMCs. Of course, an abbreviation should be only introduced if it is used later in the text.

We have now added all missing abbreviations for:

-Hepatocellular carcinoma (HCC) and microRNAs (miRNAs) in figure legend 1.

-Peripheral blood mononuclear cells (PBMCs) in section 3.1

- It is neither nice nor necessary to read “ ….by Zhao et al ….” or “…Labgaa et al..” etc. if somebody is interested in the name of the first author (s)he will find this information in the Reference list. The authors have to rephrase all these parts.

We can appreciate that our writing style could be unpleasant for some reader, therefore we rephrased the sentences and removed the credit to the first author where appropriate including:

Lines 168-169, 171-172, 341, 363, 379, 437, 449, 450, 689, 701, 711, 718, 718, 730, 731

- The authors wrote “..other phenotypic markers have been explored using different technology platforms and assays…” This is a bit too superficial and here the different technology platforms and assays must be added.

We agree with the reviewer and we indicated more specific examples with the corresponding references: “Detection of EpCAM positive CTCs with co-existing T regulatory cells (CD4+/CD25+/Foxp3+) indicated HCC recurrence [52]. In this study patients with high CTC/Tregs levels had a significantly higher risk of developing postoperative HCC recurrence than those with low CTC and Treg levels. In addition, other subtypes of CTCs have been explored, including mesenchymal CTCs the presence of which has been also associated with higher risk of tumor recurrence in HCC patients [53, 54].”

Another study shows that detection of phosphorylated ERK (pERK) and CTCs express-ing pAkt could predict response to sorafenib efficacy in advanced HCC patients, suggesting that CTCs are, similarly to tumor tissue biopsy, capable of characterizing pERK/pAkt expression [59].

We have also rephrased sentence as following: “Alternatively, other phenotypic markers have been explored, such as the hepatocyte-specific asialoglycoprotein receptor (ASGPR) [67] and the hepatocyte paraffin 1 [107] or incorporation of several markers simultaneously, as it has extensively been reviewed elsewhere [20].

Next sentence provides an indicative study of multiple marker screening; Most recently, in a prospective study of 80 HCC patients, a multimarker assay targeting together the cell-surface markers EpCAM, ASGPR and GPC3 was able to detect HCC CTCs in 97% of the patients with high accuracy; moreover a phenotypic-variant subset of CTCs was associated with aggressive disease progression and underlying metastasis, therefore highlighting the important implications CTCs may have in treatment selection [108].

- Also the statement “This group has….” must be rephrased.

We rephrased to In HCC, ISET has been utilized”

- The statement “…., etc.” is not scientific and the authors have to mention here all genes. The same must be done in line 284.

We agree with the reviewer, and we removed all the “etc” from the text.

- The authors have to mention in detail the mutations and should not simply write “..other mutations…” On top the authors have not added any citations.

We added the following text with corresponding citations studies highlighting the most frequent identified TERT and TP53 mutations as prognostic factors of poor survival [45, 47], other mutations have been described with prognostic value for HCC.” Then, the next two statements in the text describe the ‘other mutations’ as following: MLH1 mutation was specifically associated with lower survival [1], whereas another study recently demonstrated that mutations of genes from the PI3K/mTOR pathway were predictors of non-response to TKIs in patients with advanced HCC [86].”

- Under the point 3.2 one of the biggest advantages of ddPCR is missing – it is possible to measure a target without normalization to a housekeeping.

We thank the reviewer for this insightful comment, we have now added the following text with corresponding citation: In addition, ddPCR overcomes issues of normalization to housekeeping since it is based on absolute quantification of sample fractionalizing and statistics correction for multiple target molecules identified per droplet [69].” We have also updated Figure 2 with advantage included there as well.

- Under point 5 the authors have to add the aspect of tumour heterogeneity.

We thank the reviewer for addressing this important point and we have now added the following text with corresponding citations: In addition, tumor heterogeneity as a result of tumor evolution and subsequent clonal mutational differences [15] or due to treatment-induced resistant sub-clones [123] may further challenge disease monitoring and choice of therapy. These issues have prompted the research to discover other markers, exploring not only tissue but also blood samples of cancer patients [120].”

Round 2

Reviewer 3 Report

The article by Panagiota Maravelia et al. entitled “Liquid biopsy in hepatocellular carcinoma: opportunites and challenges for immunotherapy” has been improved during the revision. The current version of the manuscript can be published in the journal “Cancers”.